# Opportunity to Use New Ways of Working: Do Sectors and Organizational Characteristics Shape Employee Perceptions?

**David Giauque ***[ID]**, Frédéric Cornu** [ID]**, Karine Renard and Yves Emery**

Swiss Graduate School of Public Administration, University of Lausanne, 1015 Lausanne, Switzerland;
frederic.cornu@unil.ch (F.C.); karine.renard@unil.ch (K.R.); yves.emery@unil.ch (Y.E.)
* Correspondence: david.giauque@unil.ch; Tel.: +41-21-692-36-37

**Abstract:** The diffusion of New Ways of Working (NWW) is an important trend in contemporary organizations. Many related empirical studies have been produced, but none have focused on differences in employees' perceptions of the opportunity to use NWW according to organization sector (private, semi-public, public). This study, based on neo institutionalism and HR attributions theory, investigated these differences via a survey ($n = 2693$) of employees at private ($n = 358$), semi-public ($n = 204$), and public ($n = 2131$) organizations. Based on the use of the PLS-SEM method, as well as ANOVA tests and pairwise comparisons of marginal linear predictions, we uncover differences in perceptions between employees in different sectors regarding the possibility of using NWW. Indeed, the results show that public employees reported less opportunity to use NWW than their private and semi-public counterparts. Furthermore, private sector employees were more likely to attribute well-being and productivity benefits to NWW than their public sector and semi-public counterparts. We also show that institutional and organizational variables specific to the characteristics of organizations in the three sectors partially explain the degree of perceptions with respect to the opportunity to use NWW.

**Keywords:** New Ways of Working; sector comparison; job goal clarity; red tape; autonomy; NWW-attribution





## 1. Introduction

In the wake of the COVID-19 pandemic, New Ways of Working (NWW) became a central issue for organizations. Although the topic has been studied for many years, it gained increasing attention from both academics and practitioners during the COVID-19-related lockdowns which affected most employers and employees worldwide [1,2]. NWW may be defined as new forms of work, facilitated by information and communication technologies (ICT), that allow workers to choose when and where they work [3–5]. To summarize, NWW is a bundle of practices including teleworking, activity-based working, flexible schedules, and access to organizational knowledge. In order to define NWW more precisely, we propose the following definition taken from a recent literature review: "As part of a broader transformation of the world of work and organizations, NWW are made of practices, supported by ICT, intended to increase the flexibility, autonomy, work performance, as well as well-being of knowledge workers in their delivery of daily work, letting them choose when and where to work" [6]. NWW academic literature has been growing quickly over the last years, following organizational trends and new management tools aimed at increasing the flexibility of work conditions [7,8]. Thus, several recent studies have assessed whether these NWW benefit both employees and employers through, for example, improved individual and organizational performance [6,9–11] and employee well-being [12–14]. However, the results of these studies are mixed and inconclusive, demonstrating the need for further investigations [15].

The rapid development of NWW research and publications is of great importance, as it will enable us to assess whether these new work conditions might be helpful, as is supposed

in many pieces of literature [16,17]. Specifically, this research should be able to shed light on whether NWW can tackle new and future challenges that organizations are dealing with, such as growing socioeconomic uncertainty requiring greater agility, flexibility, and also employers' attractivity, especially for new generations looking for increased autonomy and improved work–life balance [18,19]. To date, scholars have investigated whether some of these new work conditions—i.e., using ICT to remove spatial and temporal constraints on work—benefit organizations and their employees; research has also examined whether they have any counterproductive effects, such as negative impacts on employees' psychological and social well-being [20]. Mixed empirical results have been found in terms of positive and negative outcomes of NWW implementation [5]. Nevertheless, to the best of our knowledge, no studies have tackled the question of organizational and sectoral characteristics and their potential effects on employees' opportunity to use NWW practices. Although public, private, and semi-public organizations have some similar organizational characteristics, they also differ in many important aspects. Several previous studies have pointed out that these distinct features must be considered when comparing, for instance, employees' satisfaction or perceptions of work opportunities [21,22]. In other words, organizational context matters when it comes to evaluating employees' perceptions of their ability to use NWW [23]. Thus, the main research questions of the present article are:

- Does the sector in which organizations operate matter in terms of employees' perceptions regarding their opportunity to use NWW in their organization?
- Is the perceived opportunity to use NWW practices associated with institutional and organizational characteristics?
- Do sectoral differences affect actors' attributions of the intended objectives of NWW?

This article makes several contributions. First, it answers the call of prominent NWW specialists to contextualize NWW studies with more data [24]. Second, it relies on an extensive survey comprising respondents (*n* = 2693) working in private (*n* = 358), public (*n* = 2131), and semi-public organizations (*n* = 204), allowing us to compare sectoral differences in perceptions of the opportunity to use NWW. It should be noted that our survey was carried out from October 2021 to February 2022, i.e., in a context marked by the COVID-19 crisis, but not during the lockdown period. Thirdly, in accordance with HR attributions theory, it tests whether these sectoral differences are related to employees' attributions of these new forms of work. In other words, the main purpose they attribute to their management's development and implementation of NWW could contribute to understanding their perceptions of NWW, and therefore to the real impact of NWW on employees' behavior. Finally, we also explore other sector-specific organizational features highlighted by the literature (i.e., job goal clarity, red tape, autonomy), which may further explain differences in actors' perceptions of NWW.

The remainder of the article is structured as follows: the following section introduces the theoretical foundations. The third section presents previous research on NWW to highlight what is already known with respect to its antecedents and main outcomes, looking at potential differences linked to each sector. A fourth section is dedicated to the methods, in which we present our empirical data collection as well as the methodological procedures used to test our research questions and hypotheses. Fifth, we present and discuss the results and propose some further research questions raised by our main findings.

## 2. Theoretical Backgrounds

Our article relies mainly on two theoretical orientations: sociological institutionalism and HR (human resource) attributions theory. Sociological institutionalism starts from the idea that institutions matter when it comes to explaining differences and similarities between organizations [25–27]. Organizations must be considered from an institutional perspective in the sense that they have a specific and unique history and developed around particular structures, norms, values, and cultures [28]. Organizations, seen as social institutions, should not only be perceived as structures, organigrams, or producers of rules and procedures. They are also the basis for the development of specific norms and values,

which have the particularity of partly conditioning the ways of perceiving and thinking of the different actors who evolve within them [29]. Sociological institutionalism emphasizes, moreover, that the structures, rules, routines, and functions of organizations do not only reflect functional goals, but must be read as the result of ceremonies and rituals that have become institutionalized [30,31], so that actors belonging to an organization will tend to perceive and feel situations in a rather similar way, that they will respect organizational rules and norms because any other behavior seems difficult to imagine. In other words, a logic of appropriation can be observed in organizations, underlining that the behaviors of actors, their ways of seeing and perceiving, can largely be explained by the influence of rules, norms, and organizational structures that intrude on bodies and minds [32]. In other words, the perspective of sociological institutionalism that we adopt in this article states that the actions, as well as the perceptions, of actors are largely shaped and influenced by the characteristics of their organization [33,34], and according to the job characteristics model [35], it has been largely demonstrated that specific job characteristics, which are related to organizational structure, to organizational culture, and also to implemented HR and management practices, greatly influence employee perceptions and attitudes [36–38]. This specific institutional theory highlights the importance of contextualizing organizational research and paying attention to organizational characteristics when studying employees' perceptions, attitudes, and behaviors. However, the institutional characteristics (in terms of objectives, operating rules, ethical norms, and conduct) of public, private, and semi-public organizations are not identical. We therefore postulate, in line with sociological institutionalist theory, that we will be able to identify perceptual differences concerning the opportunity to benefit from new ways of working in our research sample of public, private, and semi-public sector employees.

In the same logic of sociological institutionalism, the management tools and practices implemented in organizations must also be considered as institutional artifacts. Their specificities depend closely on the characteristics of the organizations in which they are integrated [31]. In the same way, the actors will perceive these practices and tools by being largely influenced by their organizational and institutional environment, as the sociological institutionalist perspective teaches us [28]. As a result, actors will develop perceptions of these practices and tools that owe much to the institutional environment in which they evolve. The structures, rules, procedures, and norms specific to the organization to which they belong will therefore contribute to shaping the expectations that the actors may have regarding management practices and tools. They will thus be led to interpret the reasons for these practices according to their organizational context. This theoretical position largely reflects the reflections contained in the HR attributions theory, which is why we integrate it into our reflections. The HR attributions theory stipulates that employees try to make sense of HR and management practices. In particular, they tend to make attributions to specific HR practices based on their understanding of why their organization's management has adopted them and what objectives are being pursued [39]. According to Paauwe and Boselie [40], a distinction should be made between planned, implemented, and perceived HR practices; implemented and perceived HR practices are most relevant for employees [41]. Accordingly, researchers developed a new intermediate variable: HR attributions [39,41]. Research has shown that HR attributions mediate the relationship between HR practices and HR results, and that two different kinds of attributions should be considered: HR performance attributions and HR well-being attributions [42,43].

Previous research demonstrated that well-being attributions are associated with higher levels of commitment and lower levels of job strain, whereas performance attributions are associated with higher levels of job strain [41]. This study, therefore, relies firstly on the theoretical assumption that employees make attributions regarding the intentions of HR practices, specifically those relating to NWW [43] and that these attributions may influence their perceptions of HR practices. In other words, our theoretical framework aims to explain actors' perceptions with respect to two important mechanisms. The first, inspired by sociological institutionalism, stipulates that actors' ways of seeing and perceiving are largely

dependent on the structures, norms, and rules in force within their organization. Through a process of appropriation, the behaviors of actors can be interpreted as a reflection of the institutional characteristics of their organization. The second emphasizes that the attributions and interpretations made by actors regarding management tools, largely influenced by the structures, norms, and rules in force within their organization, will have a major impact on the evaluation that actors will make of management practices and tools. In the specific case of our study, we assume that actors' perceptions working in the public, private, and semi-public sectors will differ with respect to opportunity to use NWW because the institutional characteristics of their organization are specific and shape their way of seeing and perceiving. On the other hand, in the continuity of what we have just mentioned, the actors attribute organizational intentions to the management processes and practices (for instance NWW practices) that are proposed to them. These attributions owe much to the institutional conditions in which they work, and which contribute to developing their way of seeing, perceiving, and acting.

## 3. Main Concepts and Hypotheses

### 3.1. New Ways of Working

According to recent systematic literature reviews dedicated to the NWW concept [5,44], the first articles using this notion were published in the early 2000s [45,46]; they mainly investigated flexible workspace and teleworking as facets or components of NWW and their relationships with job satisfaction. Later, during the 2010s, several other articles were published considering additional facets of NWW, namely flexible working hours and the extensive use of ICT. The relationships of these different NWW facets to important work outcomes were investigated. Overall, mixed and incongruent results have been found so far. For example, regarding NWW and employee satisfaction, some studies found a positive influence [47,48], while others found mixed effects [46,49]. Several studies noted positive impacts on work engagement [4,10]. According to previous research, NWW may also positively influence organizational attraction [50] and even organizational performance [51]. Contrastingly, other studies have identified relationships between NWW and negative work outcomes, such as decreased knowledge sharing [52], decreased work engagement and social cohesion [53], decreased productivity [46], and increased work overload [3]. As this quick literature review of NWW outcomes has shown, emergent findings are mixed and clearly point out the need for further investigations. Does the context where NWW are implemented matter?

### 3.2. New Ways of Working across Sectors: Lack of Empirical Evidence

Strikingly, the current NWW literature shows a relative lack of interest in the issue of institutional and organizational characteristics. There is no evidence so far suggesting that private organizations may be more likely to adopt NWW practices compared to their public or semi-public counterparts. Most studies on NWW are based either on samples from multiple organizations (with no investigation of the organizations' sectoral distinctions) or on single case studies. Consequently, there is a clear lack of empirical evidence allowing us to distinguish similar or divergent use and effects of NWW practices in different organizational settings—private, public, or semi-public. The first objective of our article is precisely to fill this research gap by investigating sectoral differences. Additionally, in aiming to achieve this objective, we rely on previous studies which have highlighted similarities and/or differences with respect to private, public, and semi-public organizations' characteristics and functioning. These characteristics, presented in the next section, may be useful to explain potential differences between sectors.

### 3.3. Differences between Sectors

From a broad perspective, studies focusing on organizational symbolism have consistently demonstrated that public and private organizations have different cultures, values, and objectives [54–56]. Several institutional, organizational, and job characteristics are

thought to vary across sectors. Both organizational characteristics (e.g., goal ambiguity/clarity, procedural constraints or red tape, hierarchical authority or autonomy) and job characteristics (e.g., employees' expectations towards work–life balance, social relationships and climate, and career development) are deemed to differ in public, private, and semi-public organizations [21,22,57–59]. According to this literature, public organizations are often considered to be less flexible and more formalized, procedural, and hierarchical than private or semi-public organizations. As a result, public employees have less latitude, freedom, and autonomy at work, and face more formalized and procedural constraints and strict rules [60]. The institutional and organizational characteristics, generally attributed by the scientific literature to public and semi-public organizations, more rigid, structured, and bureaucratic, compared to private organizations, will contribute to the creation of a specific culture in these organizations. Consequently, employees' expectations in terms of work opportunities within their organization will be largely influenced by the presence of these institutional and organizational characteristics. In other words, the opportunities but also the organizational constraints, which are reflected in the daily work activities of employees, contribute greatly to shaping the perceptions of the actors with regard to the work opportunities available to them [61]. In view of the institutional and organizational specificities that the scientific literature generally attributes to the various organizations in the different sectors, we can therefore expect, overall, that public employees evaluate the opportunity to use NWW less positively than private sector employees, with semi-public sector employees somewhat in the middle [21,22]. This led us to a first general hypothesis:

**Hypothesis 1 (H1).** *Public sector employees are less likely to report the opportunity to use NWW within their organization than their private and semi-public counterparts.*

We now examine the main sectoral differences by relying on the comparative literature to propose additional, more specific research hypotheses which will be tested via our empirical investigation. First, goal clarity/ambiguity is a central difference between sectors. In the scientific literature, public sector organizations are usually considered to have high levels of goal ambiguity compared to their private or semi-public counterparts [22,55,62,63]. In fact, public organizations often pursue different goals at the same time; these goals are linked to potentially incongruent or even contradictory public policies, as well as by the frequent political interventions in the implementation of public policies. Therefore, goal ambiguity is often attributed to public organizations, and has been extensively investigated by comparative scholars and even HR scholars [22,64]. The fact that public organizations and their employees deal with higher levels of goal ambiguity may impede initiatives aiming to making organizations more flexible and open to innovation and change [65]. This ambiguity of goals is not conducive to the definition of clear organizational strategies. It can also lead to opacity of objectives and working conditions [66,67]. One of the important conditions for taking advantage of the NWW is the implementation of management by objectives [68]. If the objectives appear ambiguous and unclear to employees, then the perception of being able to really benefit from and use NWW may be diminished. Thus, perceived goal ambiguity may lead employees to develop more negative or skeptical perceptions of their opportunity to use NWW. Furthermore, goal ambiguity may also influence individual attributions towards NWW practices. Hence, we hypothesized:

**Hypothesis 2a (H2a).** *Public employees are less likely to report job goal clarity compared to their private and semi-public counterparts.*

**Hypothesis 2b (H2b).** *The more employees report job goal clarity, the more likely they are to report greater opportunity to use NWW practices.*

Red tape, which relates to procedural constraints and density, is another very often diagnosed difference between sectors. Red tape is one of the few concepts native to public management literature [69]. More precisely, red tape is defined as "rules, regulations,

and procedures that remain in force and entail a compliance burden, but do not advance the legitimate purposes the rules were intended to serve" [70]. For that reason, "not all formal rules are red tape, just those that frustrate employees in achieving their goals" [71]. Bureaucratic and procedural constraints are historically well-documented organizational dysfunctions [72,73]. Individual perceptions of red tape, or bureaucratic and procedural constraints, have been found to be a predictor of numerous negative and undesirable employee attitudes, specifically in public sector organizations: intention to leave [74], lack of motivation [71], increased dissatisfaction [71,75], or even feelings of personal alienation, higher insecurity, pessimism, and mistrust [74,76]. In the scientific literature to date, procedural constraints and red tape are clearly identified as barriers to organizational change or innovation, particularly through their negative effects on actors' behavior. Red tape can therefore have a deleterious effect on actors' perceptions of the opportunities available to them in connection with NWW. Accordingly, we hypothesized:

**Hypothesis 3a (H3a).** *Public employees are more likely to report red tape compared to their private and semi-public counterparts.*

**Hypothesis 3b (H3b).** *The more employees report red tape, the less likely they are to report greater opportunity to use NWW practices.*

Finally, *autonomy* is another important sectoral difference. Public organizations must deal with political control and scrutiny, whereas private organizations are controlled by the market and economic indicators; semi-public organizations are somewhere in between [21,22]. Political accountability usually involves the development and implementation of numerous forms of governmental control [22]. The political control faced by public organizations comes with increased levels of hierarchy, and "increased levels of hierarchy are associated with many of the effects of red tape, frustrating the ability to achieve goals, and therefore might be expected to have a similarly negative effect on employee outcomes" [71]. Such hierarchies could be detrimental to the adoption of NWW. Indeed, most previous studies of private–public sector differences point out that public employees usually perceive more hierarchical control than their private counterparts [77]. Thus, public employees may feel more constrained while working; they may have the feeling they lack autonomy and freedom in their day-to-day work [78] and have less authority over tasks [58]. They may also develop distrust of management, particularly regarding the controls they face. This perception of greater control and less autonomy, or latitude in work, may resurface in public employees' perception that NWW are ultimately unavailable or unreachable. Accordingly, we hypothesized:

**Hypothesis 4a (H4a).** *Public employees are less likely to report autonomy compared to their private and semi-public counterparts.*

**Hypothesis 4b (H4b).** *The higher the level of autonomy employees report at work, the more likely they are to report greater opportunity to use NWW practices.*

In addition, due to sector-specific institutional and organizational conditions, actors are likely to make different attributions with respect to NWW practices. As is known from the literature on the development of meaning in organizations [79–81], institutional (e.g., goals, values, existing rules, regulations) and organizational (e.g., structures, task coordination, working conditions) features affect employees' expectations of their work and organization. Berger and Luckmann [82] have shown that the reality perceived by actors is a social construction. Institutional and organizational factors participate in this social construction of reality and therefore condition the actors' perceptions. In relation to Hypotheses 1–4, we assume that public sector employees are likely to have lower NWW-related expectations than their private and semi-public sector counterparts. Therefore, we believe that public sector respondents will have more neutral attributions regarding the

goals associated with NWW; both private and semi-public employees will be more likely to attribute specific organizational and strategic goals to NWW. Accordingly, we made two additional assumptions:

**Hypothesis 5a (H5a).** *Public employees are less likely to attribute well-being goals to NWW compared to their private and semi-public counterparts.*

**Hypothesis 5b (H5b).** *The more employees attribute well-being goals to NWW, the more likely they are to report greater opportunity to use NWW practices.*

**Hypothesis 6a (H6a).** *Public employees are less likely to attribute performance goals to NWW compared to their private and semi-public counterparts.*

**Hypothesis 6b (H6b).** *The more employees attribute performance goals to NWW, the more likely they are to report greater opportunity to use NWW practices.*

## 4. Methods

### 4.1. Sample

This study is based on a large-scale self-report survey. In the context of scientific research funded by the Swiss National Science Foundation, we contacted dozens of public, private, and semi-public organizations to encourage them to participate in our survey on the provision and use of NWW practices. Eleven organizations agreed to distribute our questionnaire to their employees. Five public, four private, and two semi-public organizations were involved. We also selected those organizations because we had entry points within the human resources departments of these organizations to convince them to take part in our survey. The final sample is somewhat unbalanced, insofar as we obtained more observations from public organizations ($n = 2131$) than from private ($n = 358$) or semi-public organizations ($n = 204$). This is simply because the public organizations surveyed are larger and have many more employees. Thus, the private and semi-public organizations that agreed to distribute our questionnaire to their employees are small structures (small and medium-sized organizations), whereas the public organizations participating in our survey are large cantonal or local public structures in Switzerland with several thousand employees. This imbalance between samples from public, private, and semi-public organizations can be seen as a limitation to this research. At the same time, many quantitative scientific articles have already been published with single samples, sometimes with fewer than 200 valid observations [1]. Consequently, it can be noted that this article deals with numerous empirical data, relative to three different sectors of activity, with three samples of more than 200 observations each. However, of course, we cannot consider that our research is representative of all sectors of activity and all public, private, or semi-public organizations, but this limitation is shared by most quantitative studies.

The organizations that participated in our survey are the following:

- Public sector organizations: Geneva cantonal administration; Vaud cantonal administration; Geneva city administration; Lausanne city administration; University of Lausanne.
- Private sector organizations: Intuitive (SME active in the medical field); Loyco (SME active in business consulting); Vaudoise Assurance; Romande Energie.
- Organizations from the semi-public sector (or public companies): Services Industriels Genevois (SIG); Loterie Romande.

To optimize the response rate in our online survey, we contacted the HR departments of the participating organizations, whereupon their executive members gave official approval for the study. After the test phase, a link to the online questionnaire was sent to HR departments, who invited employees to complete the survey within 3 weeks. A reminder was sent after 1.5 weeks, prompting all employees to complete the questionnaire. Because we had several participating organizations in different sectors, the collection of questionnaires was completed over a relatively long period of time (October 2021 to February 2022)

to accommodate each organization's schedule. Furthermore, to ensure complete privacy, answers were directly saved on a server belonging to our university. Thus, employees did not have access to the data, and respondents were completely and transparently informed about the research procedure. This served two purposes: increasing the participation rate and functioning as a baseline requirement to reduce common method bias [83]. To ensure that our research was ethical, we guaranteed to participating organizations that their employee data would be anonymized. We also assured them that we would not use their data for competitive or benchmarking purposes. On the other hand, we committed ourselves to providing them with an individual report on their own results, putting them into perspective in relation to the averages of all the other organizations. In this way, we guaranteed the ethical use of our data.

Out of a total of 11,675 questionnaires sent out to public organizations, 2131 valid and usable responses were received, representing a return rate of 18.25%. This modest return rate can be explained, though this is a matter of conjecture, by the fatigue of public employees regarding the questionnaires to which they are often subjected. This may be due to academic solicitations or to the proliferation of internal surveys within the organizations themselves. As for private organizations, 1101 questionnaires were sent out; 358 usable responses were received, representing a return rate of 32.51%. Finally, 693 questionnaires were sent to the two semi-public organizations participating in our survey and we obtained 204 usable questionnaires, representing a return rate of 29.43%. However, there is another limitation to our study: the fact that we cannot be certain that our samples are totally representative of the populations of the organizations studied. For reasons of confidentiality, we were unable to access information specific to each of the organizations we studied concerning the socio-demographic characteristics of their employees. This is often the case in this type of study. As a result, we cannot be certain that our data is representative of the actual populations of our organizations. That said, the samples we work with are large and can therefore be considered important.

In the overall sample of 2693 respondents, 51.5% were women, the average age was 48 years, and 49% had children in their care. The level of education was rather high: 24.9% had been in a vocational track (elementary schools to professional baccalaureate), whereas 69.8% had received higher education (college degree to university diploma). Regarding organizational tenure, 33.3% had been with their current organization up to 5 years, whereas 66.7% had been with the organization for more than 5 years.

### 4.2. Measures

All variables, items, and Cronbach's alphas (measuring internal consistency of our different variables) are presented in Table 1 and described in detail below.

### 4.3. Dependent Variable

New Ways of Working (NWW). To measure this variable, we relied on 10 items already tested in previous research [10,84]. We first conducted exploratory factor analysis to identify factors associated with the 10 items used to measure NWW; we identified five factors which correspond to five dimensions of NWW: (1) flexible scheduling, (2) flexible workplace, (3) access to information. These three variables reflect the latent variable NWW which is used in our subsequent statistical analyses. Confirmatory factor analysis of this NWW variable has been developed and displayed good fit indices (see Table 1). It should be noted that respondents were asked about their perceptions of their opportunity to use certain new work arrangements (NWW practices). Consequently, the results that we can obtain concern above all perceptions related to the opportunities offered to our respondents to use NWW practices. We do not measure actual use of NWW practices, which is difficult to do in such a study.

**Table 1.** Variables, items, and Cronbach's alphas.

| Variables: | Items Used: | Dimensions and Cronbach's Alphas |
|---|---|---|
| New Ways of Working (NWW) | Your organization offers flexible work arrangements. Please tell us whether you agree or disagree with the following proposals (1 = I do not agree; 5 = I completely agree).<br><br>1. I am free to determine my own work schedule<br>2. I am free to change my hours to choose when I start and finish my work<br>3. I am free to determine where I work, at home or at work<br>4. I am free to change where I work<br>5. I can find all the information necessary for my work on my computer, smartphone and/or tablet<br>6. I have access to all the information necessary for my work anywhere and at any time | 5 dimensions:<br><br>1. Flexible scheduling: items 1 and 2. Cronbach's Alpha = 0.86<br>2. Flexible place to work: items 3 and 4. Cronbach's Alpha = 0.72<br>3. Access to information: items 5 and 6. Cronbach's Alpha = 0.78<br>4. NWW (items 1 to 6): Cronbach's Alpha = 0.77<br><br>Results of a confirmatory factor analysis regarding the NWW variable:<br>Estimated model:<br>Chi-square: 25.801<br>Number of model parameter: 15.000<br>Number of observations: 2733.000<br>Degrees of freedom: 6.000<br>$p$ value: 0.000<br>ChiSqr/df: 4.300<br>RMSEA: 0.035<br>GFI: 0.997<br>SRMR: 0.014<br>NFI: 0.996<br>TLI: 0.992<br>CFI: 0.997 |
| Well-being attribution (WB-attribution) | Consider the flexible work arrangements implemented in your organization. What are the objectives of these arrangements? (1 = strongly disagree; 5 = strongly agree)<br>Promote the well-being of employees, making them feel valued and respected | |
| Productivity attribution (Prod-attribution) | Consider the flexible work arrangements implemented in your organization. What are the objectives of these arrangements? (1 = strongly disagree; 5 = strongly agree)<br>Increase employee productivity | |
| Job goal clarity | In relation to the demands and constraints of your work, please tell us whether you agree or disagree with the following proposals (1 = strongly disagree; 5 = strongly agree)<br>I know exactly what is expected of me<br>I know exactly what my job responsibilities are<br>I know exactly what tasks I have to perform | Cronbach's Alpha = 0.84 |
| Red tape | Some organizations have administrative rules and procedures that negatively affect their effectiveness.<br>How would you rate the degree of such rules and procedures in your organization? (1 = very low; 5 = very high). | |

**Table 1.** *Cont.*

| Variables: | Items Used: | Dimensions and Cronbach's Alphas |
|---|---|---|
| Autonomy | This section seeks to identify the main characteristics of your work, such as the level of skills required or the degree of independence. Please let us know if you agree with the following suggestions: (1 = strongly disagree; 5 = strongly agree) I take part in decisions about what my job entails I can participate in decisions that affect my work I am involved in decisions about the nature of my work I have direct influence on decisions made in my department/organization My job allows me to to take personal initiative | Cronbach's Alpha = 0.90 |

*4.4. Independent Variables*

Sector. This nominal variable has three categories (1, 2, and 3, for private, semi-public, and public organizations, respectively). This sectoral classification is based on the origin of the law that governs each organization's functioning; namely, organizations under public law are included in the public sector, while organizations under private law are included in the private sector. Semi-public organizations are either associations with a legal personality that fall under state regulation or autonomous institutions under public law with a legal personality falling under state supervision. This nominal variable is crucial when testing whether sector matters in the use of NWW.

HR attributions items. These two ordinal variables (well-being attribution (WB-attribution) and productivity attribution (prod-attribution)) were each measured by single items adapted from measurement scales previously used and tested in scientific literature (Nishii et al., 2008). The items were: "NWW aim to promote the well-being of employees, making them feel valued and respected" (WB-attribution) and "NWW aim to increase employee productivity" (prod-attribution). Both were developed specifically by the research team; they were scored on 5-point Likert-type scales from 1, strongly disagree, to 5, strongly agree.

Job goal clarity. This ordinal variable was measured via four items (scored on 5-point Likert-type scales from 1, strongly disagree, to 5, strongly agree); a sample item is: "I know exactly what is expected of me". These items have been adapted from measurement scales previously used and tested in scientific literature [85]. Cronbach's alpha for this variable was 0.82.

Red tape. A single item borrowed from Steijn and van der Voet [86] was used to measure this variable: "Some organizations have administrative rules and procedures that negatively affect their effectiveness. How would you rate the degree of such rules and procedures in your organization?" (scored on a 5-point Likert type scale from 1, very low, to 5, very high).

Autonomy. This variable was measured using five items (scored on 5-point Likert-type scales from 1, strongly disagree, to 5, strongly agree). An example item is: "My job gives me a lot of independence and freedom". These items were adapted from measurement scales previously used and tested in scientific literature [85,87]. Cronbach's alpha for this measure was 0.90.

*4.5. Statistical Analyses*

To test our different research hypotheses, we created a model with the SmartPLS 4 software, which allowed the efficient realization of partial least squares structural equation modeling (PLS-SEM). The construction of analysis models via PLS-SEM is recommended under several conditions [88–90], including the following:

(1) When researchers want to test a theoretical model from a predictive perspective.
(2) When the structural model to be tested is complex and includes several variables, indicators, and relationships between variables.
(3) When the research objective is to understand a phenomenon by exploring theoretical developments or extensions of already established theories.
(4) When the statistical model includes formative variable (NWW variable in this research).

For the above reasons, we created an analytical model to test our theoretical reflections in an exploratory manner. Figure 1, reproduced below, represents the PLS-SEM model used in our research model.

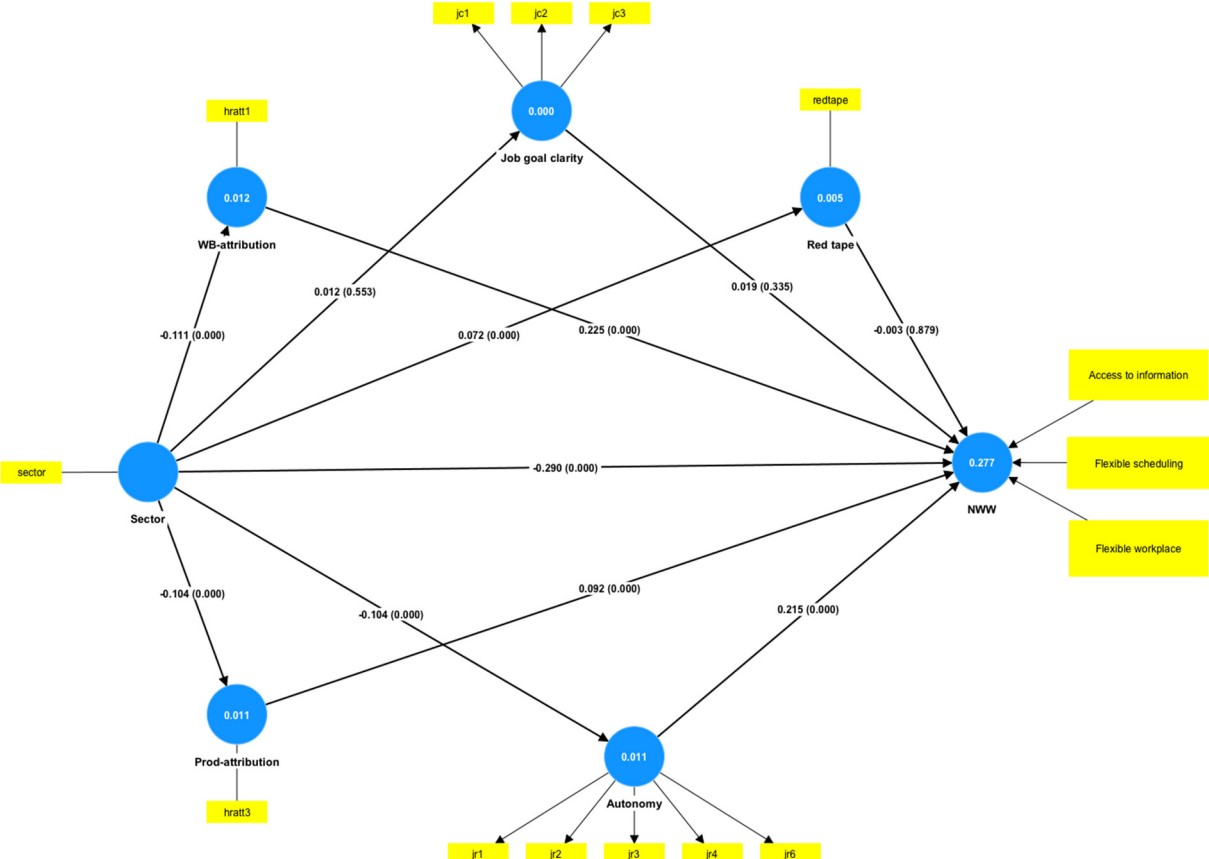

**Figure 1.** The PLS-SEM model used in our research after the bootstrapping procedure (path coefficients and *p* values between variables).

To test our model, we took several steps to ensure the normality, reliability, and validity of our data. We tested our model (via the PLS-SEM algorithm) and tested the reliability and validity of the constructs using four indicators (Cronbach's alpha, composite reliability (rho_a), composite reliability (rho_c), average variance extracted (AVE)). Overall, the variables were reliable and valid (Table 2 below). To test our reflective-formative higher-order latent variable, NWW, we performed some tests using SmartPLS 4. We complied with the usage tests recommended by the specialists [91]. All three variables correlate positively and significantly with the higher order latent variable (NWW) demonstrating the validity of this formative type dependent variable.

We then checked the discriminant nature of our analysis model, ensuring that the variables and measures included in our model were not highly related to each other. To verify the discriminative character of our model, we used the heterotrait–monotrait ratio, which is recommended [92] to assess the discriminant validity of the different constructs included in our research models. We ensured that the threshold of 0.85 was respected for each of our constructs (Table 3 below).

**Table 2.** Construct reliability and validity.

|  | Cronbach's Alpha | Composite Reliability (Rho_a) | Composite Reliability (Rho_c) | Average Variance Extracted (AVE) |
|---|---|---|---|---|
| Autonomy | 0.904 | 0.906 | 0.929 | 0.724 |
| Job goal clarity | 0.843 | 0.885 | 0.903 | 0.757 |

**Table 3.** Discriminant validity—heterotrait–monotrait ratio (HTMT).

|  | Autonomy | Job Goal Clarity | Prod-Attribution | Red Tape | Sector | WB-Attribution |
|---|---|---|---|---|---|---|
| Autonomy |  |  |  |  |  |  |
| Job goal clarity | 0.287 |  |  |  |  |  |
| Prod-attribution | 0.104 | 0.112 |  |  |  |  |
| Red tape | 0.019 | 0.073 | 0.014 |  |  |  |
| Sector | 0.110 | 0.027 | 0.104 | 0.072 |  |  |
| WB-attribution | 0.288 | 0.240 | 0.473 | 0.052 | 0.111 |  |

Finally, we also checked if our data did not suffer from multicollinearity. Hence, we checked that our variance inflation factor (VIF) indicators were below the recommended threshold of 2.5 (Table 4). We verified this for the indicators and all latent variables in our analysis model. This also indicated that our data were not affected by common method bias. Hence, we can conclude that the data are reliable and valid.

**Table 4.** Collinearity statistics (VIF)—inner model.

|  | Autonomy | Goal Clarity | NWW | Prod-Attribution | Red Tape | Sector | WB-Attribution |
|---|---|---|---|---|---|---|---|
| Autonomy |  |  | 1.144 |  |  |  |  |
| Job goal clarity |  |  | 1.110 |  |  |  |  |
| NWW |  |  |  |  |  |  |  |
| Prod-attribution |  |  | 1.296 |  |  |  |  |
| Red tape |  |  | 1.014 |  |  |  |  |
| Sector | 1.000 | 1.000 | 1.031 | 1.000 | 1.000 |  | 1.000 |
| WB-attribution |  |  | 1.418 |  |  |  |  |

The SmartPLS 4 software also offers the possibility to test whether our model has good predictive quality (can we predict our variables better than average, or better than random?). We then used the PLSpredict command to make such a predictive test. Based on this precise test, we can consider that our research model has good predictive quality with respect to the variables included, better than average or random, except for our job goal clarity variable. Furthermore, the predictive quality of the model with respect to our dependent variable NWW is high if we refer to the $Q^2$ predict value (Table 5).

In addition, we have two fit indices available for this PLS-SEM model. Both indices are indications that our statistical model fits well with our data (SRMR = 0.039; NFI = 0.926).

To obtain the necessary information to evaluate the relationships between the variables, we performed a PLS-SEM using a bootstrapping of 10,000. This method randomly generates (with replacement) subsamples from the original dataset. It is recommended to use a large number of bootstrap subsamples (at least 5000) to ensure a sufficient approximation. Finally, to obtain confirmation of our results, we conducted ANOVA tests and pairwise comparisons of marginal linear predictions including our different variables.



**Table 5.** Latent variables prediction summary—PLS-SEM.

|  | $Q^2$ **Predict** | **RMSE** | **MAE** |
|---|---|---|---|
| Autonomy | 0.010 | 0.996 | 0.792 |
| Job goal clarity | −0.001 | 1.002 | 0.773 |
| NWW | 0.119 | 0.939 | 0.766 |
| Prod-attribution | 0.010 | 0.995 | 0.819 |
| Red tape | 0.005 | 0.999 | 0.804 |
| WB-attribution | 0.012 | 0.995 | 0.804 |

## 5. Results

Table 6 in the appendices summarizes the path coefficients between the variables included in our analysis model. Thanks to this table, we can see that our sector variable is significantly and strongly correlated with our dependent variable NWW in a negative way. This means that it is primarily the respondents working in private sector organizations, then in the semi-public sector, who declare having more opportunities to use NWW. This first result confirms our general hypothesis 1.

**Table 6.** Path coefficients—Mean, STDEV, T values, *p*-values.

|  | **Original Sample (O)** | **Sample Mean (M)** | **Standard Deviation (STDEV)** | **T Statistics (\|O/STDEV\|)** | *p*-**Values** |
|---|---|---|---|---|---|
| Autonomy → NWW | 0.215 | 0.215 | 0.019 | 11.542 | 0.000 |
| Job goal clarity → NWW | 0.019 | 0.019 | 0.020 | 0.970 | 0.332 |
| Prod-attribution → NWW | 0.092 | 0.092 | 0.019 | 4.793 | 0.000 |
| Red tape → NWW | −0.003 | −0.002 | 0.017 | 0.153 | 0.878 |
| Sector → Autonomy | −0.104 | −0.105 | 0.017 | 6.023 | 0.000 |
| Sector → Job goal clarity | 0.012 | 0.012 | 0.020 | 0.602 | 0.547 |
| Sector → NWW | −0.290 | −0.290 | 0.018 | 16.273 | 0.000 |
| Sector → Prod-attribution | −0.104 | −0.104 | 0.018 | 5.634 | 0.000 |
| Sector → Red tape | 0.072 | 0.072 | 0.018 | 4.023 | 0.000 |
| Sector → WB-attribution | −0.111 | −0.111 | 0.018 | 6.270 | 0.000 |
| WB-attribution → NWW | 0.225 | 0.225 | 0.021 | 10.598 | 0.000 |

The sector is not significantly correlated with the job goal clarity, meaning that there is no statistically significant relationship between these two variables. This result does not support Hypothesis 2a. Hypothesis 2b is not supported by our data insofar as there is a positive but not statistically significant relationship between job goal clarity and NWW. Clearly, job goal clarity is not a relevant aspect in explaining our respondents' perceptions of opportunities to use NWW.

In the research model used to analyze our data, we can also see that sector is positively and significantly associated with red tape. This specific result lends credence to hypothesis 3a. This means that respondents in our samples who work in the public sector report facing more red tape than their private sector counterparts. In contrast, red tape is not statistically associated with NWW. This means that perceiving a lot or a little red tape does not have a statistically significant effect, in our data, on respondents' perceptions of the possibility of using NWW. Hypothesis 3b is therefore not supported by our empirical data.

The sector variable is also significantly and negatively correlated with the autonomy variable. This result confirms that employees working in the private sector feel more autonomy in their work than their counterparts in the public sector. This specific result supports our Hypothesis 4a. Moreover, autonomy is significantly and positively correlated with NWW, a result that is consistent with Hypothesis 4b.

If we now look at the variables related to HR-attribution, several interesting empirical findings can be made. First, we observe negative and significant correlations between sector and WB-attribution as well as prod-attribution. This means that employees in the private sector are more likely than employees in the public sector to believe that the new flexible working arrangements have the objective of promoting employee well-being and increasing productivity. These two aspects are not mutually exclusive according to our results. These results therefore support Hypotheses 5a and 6a as well. Furthermore, we find that respondents who attribute well-being and performance goals to NWW are also more likely to report higher levels of opportunity to use NWW. Hypotheses 5b and 6b are therefore supported by our empirical data.

Let us add that the test of our model via SmartPLS 4 also allows us to see if the organizational characteristics (job goal clarity, red tape, autonomy) and the NWW-attributions (WB-attribution, Prod-attribution) mediate the relationship between our sector variable and our dependent variable (NWW). As shown in Table 7, autonomy, prod-attribution, and WB-attribution statistically significantly mediate (partial mediation) the relationship between sectors and NWW. In contrast, job goal clarity and red tape do not have a statistically significant mediating effect.

**Table 7.** Specific indirect effects—Mean, STDEV, T values, *p*-values.

| | Original Sample (O) | Sample Mean (M) | Standard Deviation (STDEV) | T Statistics (\|O/STDEV\|) | *p* Values |
|---|---|---|---|---|---|
| Sector → Prod-attribution → NWW | −0.010 | −0.010 | 0.003 | 3.540 | 0.000 |
| Sector → WB-attribution → NWW | −0.025 | −0.025 | 0.005 | 5.355 | 0.000 |
| Sector → Autonomy → NWW | −0.022 | −0.023 | 0.004 | 5.221 | 0.000 |
| Sector → Job goal clarity → NWW | 0.000 | 0.000 | 0.001 | 0.386 | 0.700 |
| Sector → Red tape → NWW | −0.000 | −0.000 | 0.001 | 0.148 | 0.882 |

To complete our statistical analyses, we also conducted ANOVA tests and pairwise comparisons of marginal linear predictions including job goal clarity, red tape, and autonomy, to see whether perceptions of these variables differed significantly among public, private, and semi-public respondents, according to Hypotheses H2a, H3a, and H4a. The results confirmed our previous discussed results, and they were consistent with theoretical evidence and previous research, except for job goal clarity. Indeed, the ANOVA result showed that mean sectoral differences in job goal clarity were non-significant (Prob > F = 0.115).

Red tape is primarily an issue in public and semi-public organizations; these employees carry a heavier administrative burden than private sector employees. According to our ANOVA test, public employees reported higher levels of red tape compared to private employees, while semi-public employees reported higher levels of red tape than their private and public counterparts. Finally, as regarding autonomy, the ANOVA test showed that both public and semi-public employees were less likely to report autonomy at work than their private counterparts.

We conducted two more ANOVA tests to better investigate WB-attribution or prod-attribution. As already mentioned previously, these additional statistical tests confirm

that public sector employees were far less convinced that NWW "aim to promote the well-being of employees"; private employees—and semi-public employees to a lesser extent—expressed a different point of view. Indeed, they were more likely to attribute well-being goals to NWW practices. Differences between private and semi-public employees were not statistically significant. As for prod-attribution, the results showed that this variable was related to sectoral belonging. Public employees agreed less that NWW aim to increase employees' productivity compared to their private and semi-public counterparts. Differences between private and semi-public employees were not statistically significant.

As a conclusion to this part devoted to the empirical results, it should be noted that the variables included in our model explain 27.7% of the variance of our dependent variable NWW. This demonstrates that we capture a significant proportion of the explanation of our respondents' perception of NWW.

## 6. Discussion

The first main finding of this research relates to the comparison between private, semi-public, and public employees' perceptions regarding the opportunity to use NWW; our results showed that sector does matter in this respect. So far, previous NWW studies have mostly investigated data collected in private organizations; sector has not been considered as an explanatory variable with respect to differences in individuals' perceptions [5,44]. Our results showed that public employees were less likely to report opportunity to use NWW practices in their work environment compared to their private and semi-public counterparts. These first findings corroborate some of our theoretical expectations about sectoral differences in HR practices [93,94].

The second main finding relates to explanations of why these sectoral perceptual differences exist; according to the logic of neo institutional theory, we postulated that institutional and organizational differences could be central explanatory elements in uncovering differences in our respondents' perceptions of the opportunity to use NWW. To test this theory, we drew on previous research that highlighted institutional and organizational differences between sectors, namely red tape, job goal clarity, and autonomy at work. Our statistical analyses revealed that autonomy at work and red tape showed significant differences between respondents, particularly regarding the sector in which they work. In general, private sector employees had higher scores with respect to autonomy at work, followed by semi-public and public sector employees. In contrast, semi-public and public employees were more likely to report red tape than their private counterparts. Surprisingly, there were no differences in the means of our private, public, and semi-public sector respondents' responses regarding job goal clarity. This specific finding may be due to the fact that Swiss public administration bodies have developed several reforms inspired by the "new public management" principles. This led them to clarify their objectives and goals and measure their attainments through several qualitative and quantitative indicators [95]. Thus, except for job goal clarity, our main findings confirmed our hypotheses and supported a large body of previous research demonstrating significant differences between public, private, and semi-public employees [21,22,57–59,96]. Neo institutionalist theory underlines, indeed, that organizational origins, historical backgrounds, and culture matter when studying organizational developments and changes [27,97,98].

Consequently, a second theoretical lesson can be drawn from our data: when studying NWW, or any other form of new work, institutional and organizational characteristics must be considered. Such findings have already been made in other comparative research, notably relating to public service motivation [99–101] or comparisons of employees' work motives and attitudes [23,58,59].

A third important finding is that attributions made by actors regarding HR practices or NWW reflect to some extent the reality of institutional and organizational differences in the different sectors studied. In the case of our survey, private sector employees believed, to a greater extent than public and semi-public sector employees, that NWW promotes employees' well-being. Private sector employees also believed, to a much greater extent

than public and semi-public sector employees, that NWW aims to increase productivity. We speculate that in the private sector, increasing productivity through HR or organizational practices is considered legitimate [61]; contrastingly, in the public and semi-public sectors, perceived social impacts and public-service-oriented motives are more valued [23]. Accordingly, public sector employees were more circumspect about whether NWW is linked to improved well-being or productivity. This greater wariness is reflected in the fact that public employees' attributions regarding NWW were lower than those of their private or semi-public counterparts. These findings lend credence to HR attributions theory, which emphasizes that employees' evaluations of organizational practices depend in large part on their perceptions of why these practices have been proposed and implemented by management [41,43,102,103]. In our study, we highlight that the differences observed in terms of attributions (i.e., well-being or productivity attributions) are associated with the sectors in which our respondents worked. This element confirms the effectiveness of our theoretical framework (combining neo institutionalist and HR attributions theories) for better understanding the perceptual differences of NWW among employees working in different sectors.

Furthermore, according to our PLS-SEM analysis, it turns out that these two attributions had statistically significant positive impacts on our respondents' perceptions of the opportunity to use NWW in their own organization. This result is as expected, insofar as the opportunity to use NWW practices is the condition for the development of individual opinions about the main objectives of these NWW. It is therefore quite normal that there is a strong correlation between the perception of the opportunity to use NWW and attributions in terms of well-being or productivity. This highlights that it is probably necessary to look at the values that are disseminated and considered legitimate within organizational cultures and among individual employees [104,105]. This also indicates that individual attributions may have a direct impact on perceptions of the opportunity to use NWW. In contrast, it is more surprising to find that both types of attributions had a positive and statistically significant effect on NWW perceptions; one might have imagined that attribution in terms of productivity could be negatively correlated with respondents' perceptions of NWW. This suggests that it is worth studying other HR attributions to see whether they are positively or negatively linked with the opportunity to use NWW.

The final important contribution of our paper is to demonstrate that specific organizational characteristics (in our case perceived autonomy) and NWW attributions have mediating effects between the sector and actors' perceptions of their opportunity to use NWW. This suggests that, among other factors, organizational communication, which delivers a consistent message around the implementation of NWW, can contribute to creating favorable or unfavorable perceptions regarding the opportunity to use NWW [106].

*Limitations and Future Research*

As with all quantitative research based on data from one questionnaire, it is impossible to draw definitive conclusions about the causal relationships between our different variables; this is the most well-known limitation of this type of survey. Therefore, we speak of significant relationships, correlations, and associations and not of causal effects.

Concrete work activities performed by actors vary greatly according to their job description; the specificities of different tasks performed and associated professions can therefore also shape actors' perceptions of HR practices and of NWW. We did not include any job-related variables in our study. By crossing the sector variable with job-related variables, it would be possible to get an even more precise idea of the perceptions of actors and identify clusters in relation to the perceptions of opportunities for using NWW practices going beyond sectoral borders. Such research would be an undeniably significant addition to the understanding of sectoral differences.

Another important methodological issue relates to our one-sided methodology (i.e., a self-report survey to collect predictor and outcome variables), which can result in common method biases (Podsakoff et al. 2003). This strategy may inflate the reported effect sizes.

To check our data did not suffer from these biases, we performed Harman's single factor test, showing that all the variables in our model account for only 16.3% of a single factor, i.e., our data are free of common method biases. We also tried to minimize this problem through the conditions of the survey.

Finally, other HR practices (such as recruitment and selection, training and development, compensation, etc.) were not investigated in our survey. These practices may be aligned with the introduction of NWW practices, leading to a potentially increased HR "bundle effect" on employees' productivity and well-being [107,108]. In addition, management styles can have more or less positive effects on the development and use of NWW in organizations. Studying different management styles (management by objectives; management by indicators; management by processes; etc.) and their impacts could lead to interesting results. Similarly, organizational cultures may have effects on the development and use of NWW. Studying these relationships between organizational cultures and NWW, beyond sectors, is an important issue to consider in future research.

## 7. Conclusions

The purpose of this study was to further investigate the issue of NWW and related perceptions of employees working in different sectors (public, private, and semi-public). Based on theoretical foundations related to neo institutionalism and HR attributions theory, we hypothesized that our respondents' perceptions of NWW would be associated with the sector in which they work, the institutional and organizational characteristics of the organization to which they belong, and the attributions they made of the objectives underlying the development and implementation of NWW in their organization. We have thus been able to show that there are indeed differences in perceptions of NWW according to organizational sector, but also according to the attributions made by actors. Moreover, these attributions largely overlap with sectoral boundaries in our study. This study makes several original contributions to the literature on NWW. It is first to compare sectors and to investigate perceptual differences of NWW with respect to employees in three sectors (private, public, semi-public). Second, it tests institutional and organizational variables to see if they contribute to explaining the perceptual differences uncovered in our survey. Finally, it highlights the importance of NWW attributions in the formation of actors' perceptions. Further research is needed on additional variables which may influence employees' behavior and outcomes related to the introduction of NWW practices. Aspects linked to the particularities of the work performed by employees, as well as aspects in terms of organizational culture or organizational climate, could be the subject of further investigation. The question of leadership within organizations may also constitute an interesting new avenue of research, to be considered in future studies of actors' perceptions of the opportunity of using NWW.

**Author Contributions:** Conceptualization, D.G. and Y.E.; Methodology, Y.E.; Software, D.G., F.C. and K.R.; Investigation, D.G.; Resources, F.C. and K.R.; Writing—original draft, D.G.; Writing—review & editing, D.G., F.C., K.R. and Y.E.; Supervision, D.G. All authors have read and agreed to the published version of the manuscript.

**Funding:** This research was funded by the Swiss National Science Foundation, grant number 100018_185133.

**Institutional Review Board Statement:** Ethical review and approval were waived for this study because each of the organizations studied gave their explicit agreement to take part in our survey. In addition, the conditions of respondent anonymity and data confidentiality were respected at every stage of our survey.

**Informed Consent Statement:** Informed consent was obtained from all subjects involved in the study.

**Data Availability Statement:** Our empirical data are available publicly: https://www.swissubase.ch/fr/ (accessed on 30 May 2023).

**Conflicts of Interest:** The authors declare no conflict of interest. The funders had no role in the design of the study; in the collection, analyses, or interpretation of data; in the writing of the manuscript; or in the decision to publish the results.

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
