# Peer review of "Opportunity to Use New Ways of Working: Do Sectors and Organizational Characteristics Shape Employee Perceptions?"

_sustainability, doi:10.3390/su151411167_

Round 1
Reviewer 1 Report
Dear authors,
It is an honor for me to review your article.
The article presents an interesting theme with international impact.
In this regard I have to mention some aspects to improve:
1. In line 10 there is a missing r.
2. In the introduction, they begin by talking about the COVID-19 pandemic, however, they use 2 references from 2016 and 2012, respectively. This is an inconsistency. Was there a similar pandemic in 2012 and 2016? It must be taken into account that there must be coherence between the text and the theoretical support used. In this sense, the introduction must be redone and, if it is framed in a post-pandemic scenario, citations from 2020 to 2023 must be used.
Lines 60 to 68 refer to the research problem and research questions, therefore, the research objectives. This must be located at the end of the theoretical framework or within the methodology.
At the methodological level, they present various methodological doubts. First, the type of sampling, sample selection criteria, or sampling error are not specified.
In line 388 they reflect "Appendix" and the table is below. This is not understood, are they going to put it in an appendix or include it in the text?
Likewise, they do not reflect the investigation procedure or the ethical code followed in the investigation.
Author Response
Thank you very much for your review of our article and for the valuable comments. You'll find answers to your comments below.
- Line 10, the r has been erased.
- - We revised the introduction to include additional references in line 31 to two articles on teleworking in the context of the COVID-19 crisis. This is to eliminate the ambiguity you mention in your second point. Lines 73-75 have been added to clarify that our field study took place between October 2021 and February 2022, during a period marked by the COVID-19 crisis, but outside a lockdown period.
- We have also added a note on the limitations of our research samples. This note can be found on lines 381-388.
- n line 388 they reflect "Appendix" and the table is below. This is not understood, are they going to put it in an appendix or include it in the text? -> you are completely right we have deleted the reference to the appendix on line 397.
- Lines 372-377 have been added in response to your last comment about our code of ethics and conduct regarding the use of collected data.
We hope to have responded to your comments to your satisfaction. Thank you once again for your detailed reading and your remarks.
Reviewer 2 Report
Introduction:
In the introduction, a scientific definition of NWW is needed.
Line 40 - The statement "are mixed and inconclusive, demonstrating the need for further investigations" is too strong. Please, present some literature to demonstrate this sentence.
Line 42-44. Please, provide some references.
Line 45-53. The literature is not well described. Please, provide more details.
Line 54 - the effect on the development and opportunity .... for who? Who is the stakeholder involved?
Lines 59-60. Why only employees' satisfaction or perceptions of work opportunities? This selection should be justified. There are different type of social effects related to working conditions.
Line 60 . Please, add a reference.
Lines 61-67. the research questions should be better formulated.
Lines 70-73. the sample taken for each type of organisation should be homogeneous.
Line 92. Please, explain HR.
Lines 123-132 Add some references.
Line 139; 142 - the references are not in the Journal format.
Section 3:
Did you conduct a literature review? Maybe it is a state-of-the-art or theoretical background. Please, modify accordingly the title of section 3. Moreover, subsections interrupt the flow of reading. I suggest removing them and re-organizing the content (also with respect to the previous comment). Indeed, it is too long and it is not clear the final aim of this section.
Section 4.- method
The method is not well described and presented. Section 4 missed a description of the methodological approaches (methods, tools used) used for conducting this study. Please, provide it also with some diagram. For example: a description of Cronbach'Alfa is needed. You never mention it before now (line 388). What was its role? Please, explain its role and how it works. The same is for PLS-SEM (presented in line 441).
Line 334: how did you select the sample?
Line 344: How did you overcome this limitation?
Line 349: Which quantitative studies?
A description of Cronbach'Alfa is needed. You never mention it before now. What was its role? Please, it should be explained in the introduction of the method section.
Table1: how did you select the variables to be considered? Please, explain and justify them.
Results:
The results are not well presented. I suggest including some figures (and not tables - are already enough), for improving the readiness of the manuscript.
The conclusions are not sufficiently treated. Please improve this part by including further implications of the work.
Author Response
First of all, thank you for your precise reading and for your precious remarks and comments. We tackled all of them. Below you will find your remarks and -> our answers:
- According to your suggestion we have defined the notion of NWW more precisely (see lines 35-40)
- We added a reference to justify our statement "are mixed and inconclusive, demonstrating the need for further investigations" (see line 46).
- Line 42-44. Please, provide some references -> we provided some new references (see line 49).
- Line 45-53. The literature is not well described. Please, provide more details -> we provided detailed description of the literature regarding NWW and teleworking in the chapter 3. Therefore, so as to avoid repetition we didn't mention other references in the introduction.
- Line 54 - the effect on the development and opportunity .... for who? Who is the stakeholder involved? -> our original sentence was: Nevertheless, to the best of our knowledge, no studies have tackled the question of organizational and sectoral characteristics and their potential effects on the development and opportunity to use NWW practices. We decided to transform this sentence into: Nevertheless, to the best of our knowledge, no studies have tackled the question of organizational and sectoral characteristics and their potential effects on employees' opportunity to use NWW practices (see line 60).
- Lines 59-60. Why only employees' satisfaction or perceptions of work opportunities? This selection should be justified. There are different type of social effects related to working conditions -> Yes, we totally agree with you working conditions are related to different social effects as well. But due to our research objectives, our article deals with the degree to which they perceive they have the opportunity to use NWW in their day-to-day work. We could have mentioned other work outcomes of course, but the latter would not be related to our article objectives.
- Line 60 . Please, add a reference -> we added a reference (see line 66)
- Lines 61-67. the research questions should be better formulated -> According to the fact that this remark is unique among the seven reviewers, we decided to keep our research questions as they are. In our opinion, they clearly reflect the main questions of our article.
- Lines 70-73. the sample taken for each type of organisation should be homogeneous -> yes, we agree with you, usually the sample must be harmonized in such comparative analysis. Nevertheless, according to the fact that the size of participating organizations was very different it is more accurate to develop our statistical analyses on our collected data and not on homogeneous sample sizes, which could be a superficial technique to address the issue of having different samples sizes among sectors studied.
- Line 92. Please, explain HR -> HR for Human Resource (see line 99).
- Lines 123-132 Add some references -> some references have been added in the new version (see lines 133 and 135 for instance).
- Line 139; 142 - the references are not in the Journal format -> thank you for your attention. The correct citation format has been used for these references.
- Did you conduct a literature review? Maybe it is a state-of-the-art or theoretical background. Please, modify accordingly the title of section 3. Moreover, subsections interrupt the flow of reading. I suggest removing them and re-organizing the content (also with respect to the previous comment). Indeed, it is too long and it is not clear the final aim of this section -> you are right, this section is not a literature review per se, but a theoretical background. Therefore, we modified the title of this section. This section is long, yes, but we need all these explanations so as to justify our research hypotheses. Otherwise, we will have some issues with other reviewers.
- The method is not well described and presented. Section 4 missed a description of the methodological approaches (methods, tools used) used for conducting this study. Please, provide it also with some diagram. For example: a description of Cronbach'Alfa is needed. You never mention it before now (line 388). What was its role? Please, explain its role and how it works. The same is for PLS-SEM (presented in line 441) -> We added a very short and brief description of Cronbach's Alpha (line 407). According to the fact that this internal consistency measure is very well-known, we will not specify further the importance of the Cronbach's Alpha. And PLS-SEM is the abbreviation of partial least squares structural equation modeling (PLS-SEM) (see lines459-469 for further explanations regarding this specific statistical method). In our view, this is no need to specify further this method otherwise the paper will be transformed into a methodological article, which is not our objective.
- Line 334: how did you select the sample? -> simply because the organizations involved in our research have accepted our invitation to participate. But also because we had entry points within the human resources departments of these organizations to convince them to take part in our survey. We added this point in the new version of the article (see lines 342-344).
- Line 349: Which quantitative studies? -> We added a reference with respect to this specific point (see line 354).
- Table1: how did you select the variables to be considered? Please, explain and justify them -> all the variables considered in our research are explained in detail after the presentation of Table 1. According to the fact that we mainly rely on already tested scales we mentioned the references as well in this variables' description part.
- The results are not well presented. I suggest including some figures (and not tables - are already enough), for improving the readiness of the manuscript -> Yes, figures could be helpful to read the article but the results are presented mainly with two tables and one figure. Thus, this seems OK this way.
- The conclusions are not sufficiently treated. Please improve this part by including further implications of the work -> We add some new research avenues at the end of our conclusion (see lines 732-737).
Reviewer 3 Report
Just a few editorial remarks:
Lines 452-453
Figure 1, reproduced below, represents the PLS-SEM model used in our research model.
---
IMHO: Figure location is too far from its actual invocation in the text!
Lines 467-488
---
IMHO the actual tables: from Table 3 to Table 5 are lacking actual reference in the text, here!
Author Response
Thank you for your reading and for your comments, which have helped us to improve our article. We have taken your comments into account in the new version of our text.
Your comment: Figure 1, reproduced below, represents the PLS-SEM model used in our research model.
---
IMHO: Figure location is too far from its actual invocation in the text!
Our answer: you are probably right, but the Figure 1 is also used to show results. Thus the location is the right one, the other option would be to duplicate the figure, which is not a good solution.
Your comment:
Lines 467-488
---
IMHO the actual tables: from Table 3 to Table 5 are lacking actual reference in the text, here!
Our answer: we included references of the tables directly in the text now.
Reviewer 4 Report
The research is quite interesting, a non-standard analytical method using the PLS-SEM model is chosen. I consider the questionnaire survey to be sufficient for the chosen method and form of research. The mathematical and statistical background of the research is supported in tables and explained in the text.
I consider the expressed hypotheses to be too general and predictable in their essence.
I would recommend the authors to describe in more detail the model presented in Figure 1 and to confront its structure in a discussion with the conclusions of other authors.
Author Response
Thank you for reading and for your comments, which have helped us to improve our article. We have taken your comments into account in the new version of our text.
Your comments:
The research is quite interesting, a non-standard analytical method using the PLS-SEM model is chosen. I consider the questionnaire survey to be sufficient for the chosen method and form of research. The mathematical and statistical background of the research is supported in tables and explained in the text.
I consider the expressed hypotheses to be too general and predictable in their essence.
I would recommend the authors to describe in more detail the model presented in Figure 1 and to confront its structure in a discussion with the conclusions of other authors.
Thank you for your suggestions. Which mainly imply that we change all our research model so as to evoke other research hypotheses. We choose to keep our hypotheses as they were in our version because other reviewers (total of 7 reviewers!) considered them as non problematic.
The model presented in Figure 1 represents the relationships between our different variables. It is our model and this model is an original one. We could discuss this model with respect to other articles (could be useful) but this could be another article (our article is very long at this stage).
But your suggestions are very interesting and useful. In our specific case they imply simply too much changes and the writing of another article, fast completely different.
Reviewer 5 Report
I carefully read the 23-page manuscript, the thematic focus of which is very close to me, it can clearly be judged from its content that it is the result of a time-consuming scientific work that also bore fruit.
However, in the abstract, I would recommend paying more attention (at least two sentences) to the scientific research methods used.
In my opinion, the part "2. Theoretical backgrounds" should be supplemented with several Central European sources devoted to the investigated issue, which would give the authors a broader view of the investigated issue of human resources. These are works such as:
Peracek T. (2020). Human resources and their remuneration: managerial and legal background. Paper presented at RELIK 2020: Reproduction of human capital - mutual links and connections, Prague, Czech Republic, November 5-6, 2020, pp. 454-465
Gheorghe, M. (2020). CONSIDERATIONS ON THE MEANING OF THE NOTION OF "WORKING TIME" IN THE LIGHT OF RECENT C.J.E.U. JURISPRUDENCE. Perspectives of Law and Public Administration, 9, (1) pp. 58-64.
Hard to say which part is the most valuable, but probably the results and recommendations.
I am convinced that the authors will incorporate the mentioned changes into the text, which will increase its value.
Author Response
Thank you for reading and for your comments, which have helped us to improve our article. We have taken your comments into account in the new version of our text.
Your comment:
However, in the abstract, I would recommend paying more attention (at least two sentences) to the scientific research methods used.
Our answer:
Yes, you are right. We included two more sentences in the abstract to reflect better the methods used (see lines 17-19).
Your comment:
In my opinion, the part "2. Theoretical backgrounds" should be supplemented with several Central European sources devoted to the investigated issue, which would give the authors a broader view of the investigated issue of human resources. These are works such as:
Peracek T. (2020). Human resources and their remuneration: managerial and legal background. Paper presented at RELIK 2020: Reproduction of human capital - mutual links and connections, Prague, Czech Republic, November 5-6, 2020, pp. 454-465
Gheorghe, M. (2020). CONSIDERATIONS ON THE MEANING OF THE NOTION OF "WORKING TIME" IN THE LIGHT OF RECENT C.J.E.U. JURISPRUDENCE. Perspectives of Law and Public Administration, 9, (1) pp. 58-64.
Our answer:
Thank you for your suggestions regarding these new references. They are interesting for sure but they are not entirely in the scope of our article. It is why we decided not to include them in our references.
But I do agree that Central European sources could also be integrated better in literature review in the future. Thank you for your understanding.
Reviewer 6 Report
I have no critical notes other than the citation of the references in the text. There is used two types - see lines 138-145 (I did not find Paauwe and Boselie (2005) in the references,) and 678.
Author Response
Thank you for your reading and comments.
Your comment:
I have no critical notes other than the citation of the references in the text. There is used two types - see lines 138-145 (I did not find Paauwe and Boselie (2005) in the references,) and 678.
Our answer:
Yes, you are right. We've corrected this in the new version.
Reviewer 7 Report
Dear,
the research is interesting and current, today when an increasing number of corporations work remotely.
The research is well structured and adequately presented. In some segments, more explanations are missing and the work would be even better.
- Explain HR attributions theory. What is HR ("HR practices", "HR results", "HR performance", "HR well being"...). An explanation of what HR is missing.
- Typographical errors appear in several places.
- On page 10 in line 396, five dimensions of NWW are mentioned, and the writing is three. What are the others?
- Explain "Heterotrait-Monotrait ratio".
- In conclusion, add what other research would clarify this issue.
All the best
Author Response
Thank you very much for your review of our article and for the valuable comments. You'll find answers to your comments below.
Your comment:
- Explain HR attributions theory. What is HR ("HR practices", "HR results", "HR performance", "HR well being"...). An explanation of what HR is missing.
Our answer:
We explained this in the new version. HR means Human Resource. This expression is related to a literature stream which is very specific (see line 102).
Your comment:
- Typographical errors appear in several places.
Our answer:
We corrected all these typographical errors when we identified them through our multiple readings.
Your comment:
- On page 10 in line 396, five dimensions of NWW are mentioned, and the writing is three. What are the others?
Our answer:
It was an error. Thank you for having identified it. Our exploratory factor analysis leads us to identify three factors, not five, which correspond to three dimensions of our NWW variable. We corrected this error in the new version of the article.
Your comment:
Explain "Heterotrait-Monotrait ratio".
Our answer:
We explained this notion better and included as well a new reference with respect to this Heterotrait-Monotrait ratio (see lines 495-496).
Your comment:
- In conclusion, add what other research would clarify this issue.
Our answer:
We completed our conclusion with some new research avenues with respect to the issue tackled in our article (see lines 737-742). But, of course, other research ideas could be mentioned but we have to keep in mind the length of our article. It is why we completed the conclusion but briefly.

Round 2
Reviewer 1 Report
thanks for applying my suggestions